# Challenges in Predicting the Change in the Cumulative Exposure of New Tobacco and Related Products Based on Emissions and Toxicity Dose–Response Data

**DOI:** 10.3390/ijerph191710528

**Published:** 2022-08-24

**Authors:** Yvonne C. M. Staal, Wieneke Bil, Bas G. H. Bokkers, Lya G. Soeteman-Hernández, W. Edryd Stephens, Reinskje Talhout

**Affiliations:** 1National Institute for Public Health and the Environment (RIVM), P.O. Box 1, 3720 BA Bilthoven, The Netherlands; 2School of Earth & Environmental Sciences, University of St Andrews, St Andrews KY16 9AJ, UK

**Keywords:** tobacco products, carcinogenicity, cardiovascular diseases, heated tobacco, cumulative exposure, relative potency

## Abstract

Many novel tobacco products have been developed in recent years. Although many may emit lower levels of several toxicants, their risk in the long term remains unclear. We previously published a method for the exposure assessment of mixtures that can be used to compare the changes in cumulative exposure to carcinogens among tobacco products. While further developing this method by including more carcinogens or to explore its application to non-cancer endpoints, we encountered a lack of data that are required for better-substantiated conclusions regarding differences in exposure between products. In this special communication, we argue the case for more data on adverse health effects, as well as more data on the composition of the emissions from tobacco products. Such information can be used to identify significant changes in relevance to health using the cumulative exposure method with different products and to substantiate regulatory decisions.

## 1. Introduction

In recent years, manufacturers have developed and marketed several new types of tobacco and related products (TRPs), such as electronic cigarettes, heated tobacco products (HTPs), and oral nicotine products [1]. When novel TRPs become available to consumers, the risk of developing use-related diseases, especially in the long term, is largely unknown. Still, many of these products are marketed and perceived as reduced-risk products [2,3,4,5]. Indeed, novel TRPs generally emit lower amounts and concentrations of known carcinogens and other types of tobacco-related toxicants [6,7]. However, these lower concentrations do not necessarily translate to commensurate reductions in risk, since, at high concentrations, the S-shaped dose–response curves flatten, implying that a lower concentration does not lead to a proportionally lower risk.

As a large part of the composition of new TRP emission overlaps with cigarette smoke, some diseases likely to arise after long-term use are tobacco-related, but these new TRPs may also cause different toxicities due to compounds not present in tobacco smoke. For example, in HTP emissions, other hazardous chemicals have been identified [8].

Most tobacco-related diseases appear only after several decades of use [9]. However, to inform new regulations and substantiate regulatory decisions that aim to protect public health, it is crucial to have at least an estimate of the risk of new TRP use before they become available on the market. A (quantitative) estimate of the change in exposure can inform a change in health impacts. Based on such information, policymakers worldwide, in line with their national regulations, may regulate the product by banning it or by setting limits to specific contents or emissions [10]. In this respect, there is a difference between the Tobacco Products Directive (TPD, 2014/40/EU) and other regulatory frameworks, such as the Registration Evaluation Authorisation and Restriction of Chemicals Regulation (REACH, (EC) No. 1272/2008), with regard to the information that needs to be provided. The toxicological and exposure information that a registrant is requested to provide according to the REACH regulation is extensive compared with the available scientific information that needs to be submitted on novel TRPs according to the TPD Article 19.1a. EU member states may request additional information according to the TPD, but this is not harmonized and, in practice, the available information to regulate TRP products is relatively limited.

To evaluate the health impact of novel TRPs as compared with conventional cigarettes, it is important to study the total chemical burden in the emissions of new tobacco and related products, rather than focusing on isolated components [11]. Previously, Slob et al. developed and published a model that allowed us to make an assessment of the expected change in cumulative exposure (CCE) based on eight compounds found in the emissions of conventional cigarettes and HTPs [12]. The assessment underlying this CCE methodology is based on the principle of dose-addition [13], which adds the doses of the compounds involved after adjusting each dose by the relative potency factor (RPF) of each compound. This comparative method enables the assessment of reduced risk claims, an option legally available to industry in, for example, the U.S. [14].

After the publication of Slob et al. [12], we investigated the possibilities of improving the estimate of the CCE by using additional data regarding cancer. In addition, we explored the possibilities of using data regarding non-cancer endpoints. Our attempt to find more carcinogenic compounds with relevant data resulted in the identification of various issues hindering the continuation of this work, due, mainly, to a lack of quantitative emission data or relevant dose–response data. We also came across such issues during our research on potentially useful noncancer data, but also found more fundamental issues. The current paper reflects on several challenges we encountered in predicting the cumulative exposure to substances in TRP emissions. As long as these challenges are not tackled, it will not be possible to provide scientists and regulators with the information needed to assess any health claims made for novel TRP.

## 2. Extending the CCE Method for Carcinogenicity

The higher the fraction of the total number of compounds that can be used to determine the carcinogenic potency of a mixture, the greater the reliability of the predicted carcinogenic health impact of TRP use. Our previous paper described a method for eight carcinogenic compounds that are present in the emissions of HTPs at lower quantities than in tobacco smoke [12]. For these particular eight compounds, the CCE was found to be substantial, which would imply a substantial mitigation of health effects. However, it does not seem likely that these eight compounds are representative of the hundreds of carcinogenic compounds in the emissions. The chemical analysis to assess the emissions of HTPs focused on compounds that are included on lists, such as the PMI-58 list (this consists of the 58 compounds reported by Phillip Morris International) [8] or the list of the U.S. Food and Drug Administration (FDA) on hazardous or potentially hazardous compounds (HPHC) [15], which find their foundation in the evaluation of compounds in the emissions of cigarette smoke. Therefore the determination of hazardous chemicals in new TRP emissions is biased toward chemicals with high emission values in cigarette smoke.

By incorporating compounds with higher concentrations in HTP emissions and by including the compounds not necessarily detected in conventional tobacco smoke, the assessment would reflect a more balanced estimation of risk. Therefore, non-targeted analysis (pre-selection of compounds based on emissions) should be performed prior to targeted analysis (measurement of the emissions of selected compounds) to also include compounds not necessarily present in the emissions of the conventional cigarette. Moreover, targeted analysis should rely on replicated emission measurements to reliably determine the measurement error. Such targeted analysis also should include a reference product (tobacco cigarette) as a positive control. Emission measurements in the scientific literature (summarized in Simonavicius et al. [16]) do not always include data on a comparator product or use different comparator products (different cigarettes), which limits the ability to compare emission data between products.

So far, four carcinogenic compounds were found in iQOS emissions at higher quantities than in tobacco smoke: 3-chloro-1,2-propanediol, furfural, glycidol, and 2-furanmethanol [8]. Therefore, including these four compounds in the calculation of the CCE may give a different picture to that concluded so far [12]. Clearly, emission data for both cigarettes and HTPs would be needed for these four compounds. While this is the case, the data are un-replicated measurements, which means that information regarding the uncertainty in the measured emissions is unavailable [8,17]. Without such information, the crucial evaluation of the uncertainty in the CCE is, strictly speaking, not possible. Furthermore, for three of the four newly identified compounds, no carcinogenicity data via inhalation are available, therefore impairing the derivation of suitable RPFs (Table 1).

Data are needed on the carcinogenic potency of compounds via the inhalation route in a standardized laboratory test protocol to obtain RPFs. Because absolute potency measures, such as the No-Observed Adverse Effect Level (NOAEL) or Bench-Mark Dose (BMD), may depend on factors such as exposure duration, exposure route, sex, species, strain, and life stage, RPFs would ideally be estimated based on studies where such factors are similar. Currently, the available carcinogenicity data for the compounds in TRP emissions relate to chronic cancer studies, where exposure duration is often the same (two years). Such carcinogenicity data are not available for all compounds in TRP, implying that carcinogenicity studies for these specific compounds via the inhalation route are not required by any non-tobacco legislation. Moreover, an issue with some of the available carcinogenicity data is the dosing, resulting in the highest dose(s) not causing any sign of toxicity and therefore providing no relevant dose–response information [19].

The recommendations for better data are summarized in the text box below (Box 1).

Box 1Summary of the problem and recommendations to extend the CCE method for carcinogenicity.**Problem:** the CCE methodology could not be applied to an extended database so as to provide better information on the relative carcinogenicity of a novel TRP due to the lack of robust carcinogenicity data for the inhalation route and a lack of data on the composition and concentrations of compounds in their emissions.
**Recommendations:**

•Non-targeted screening studies on the composition of emissions need to be conducted for novel TRPs to obtain information on the compounds and their quantities in these emissions;•After selecting compounds from the non-targeted studies based on quantity and hazard, harmonized and replicated measurements of the concentrations of TRP emissions need to be conducted, in tandem with those in a reference cigarette, to allow reliable comparison;•More carcinogenicity data via the inhalation route are required for compounds present in novel TRPs and/or in cigarettes, especially for chemicals with high concentrations in emissions in either of them. Data from 2-year carcinogenicity studies could serve this need, but other in vivo dose–response data could also be used if the endpoint can be considered predictive of cancer, such as micronucleus effects [20];•Efforts should be made to investigate how the outcomes of such in vivo assays relate to their in vitro equivalents (e.g., [21]), and whether in vitro assays could also provide approximate estimates of the relative potencies, possibly adjusted by information on kinetics and dynamics. Such in vitro information could be obtained for less effort and cost, and is preferred from an ethical viewpoint compared with in vivo carcinogenicity or micronucleus studies.


## 3. CCE Method for Non-Carcinogenicity

The use of tobacco products is also associated with an array of non-cancer diseases, such as cardiovascular diseases, diabetes, chronic obstructive pulmonary disease (COPD), and fertility problems [22]. If the CCE method based on carcinogenicity could be adapted for these diseases, it would allow us to qualitatively compare the impact of different tobacco products on the development of other adverse health effects in addition to carcinogenicity, and therefore provide a more complete estimate of the health implications associated with changes in cumulative exposure.

Cardiovascular diseases (i.e., atherosclerosis, coronary heart disease, stroke, peripheral arterial disease, and abdominal aortic aneurism) are leading causes of death worldwide [23], and are positively associated with tobacco use, resulting in higher mortality rates, shorter lifespans, and lower quality of life among users compared with non-users of tobacco products [24,25]. Thirteen compounds in tobacco smoke, part of the FDA HPHC list, are established cardiovascular toxicants: acrolein, arsenic, benzo[a]anthracene, benzene, benzo[b]fluoranthene, benzo[k]fluoranthene, benzo[a]pyrene, chrysene, cobalt, hydrogen cyanide, lead, phenol, and propionaldehyde [15].

Suitable controlled disease models are available for studying the development of atherosclerosis; therefore, this effect was chosen as the first non-cancer disease associated with tobacco use to be explored [26]. Because of profound differences in lipid homeostasis between humans and mice, genetically engineered mouse models have been developed (e.g., ApoE -/- knockout model) to alter the disease development to resemble human lesions [27]. Unfortunately, to date, only acrolein [28], arsenic [29,30,31,32,33,34], benzo[a]pyrene [35,36,37,38,39], and nicotine [40] have been tested in this transgenic mouse model. We considered this dataset too limited to be sufficient for the derivation of RPFs to estimate the CCE, as it should be representative of the mixture of compounds.

We then considered parameters associated with cardiovascular disease in standardized testing protocols with wild-type rodents (i.e., the OECD TG 413 and TOX/TR 90-day study by the US national toxicology program (NTP)), despite the species differences mentioned above. Compounds in tobacco emissions tested by the NTP (*n* = 36) are provided in Appendix A. The parameters considered were the heart weight, pathology of the heart and vascular system, blood cholesterol and triglycerides, and white blood cell parameters. However, none of the studies contained cholesterol as a measured parameter and none of the other parameters showed a statistically significant trend.

As a final attempt to derive RPFs for non-carcinogenic adverse health effects, we observed the four parameters most regularly observed in the NTP 90-d study protocol (body weight (bw) and relative (to bw) weights of the liver, kidney, and thymus), hypothesizing that any effect in one or more of these parameters would reflect the systemic toxicity of the compound. This analysis successfully provided RPFs for bw and liver weight in mice per sex exposed via the inhalation route (Appendix A). However, as for the carcinogenicity endpoint, the number of RPFs to consider for CCE (five compounds for bw changes and seven compounds for liver effects) is only a small fraction of the total number of over 7000 compounds identified in tobacco smoke [22]. Recommendations for non-carcinogenicity data are summarized in Box 2.

Box 2Summary of the problem and recommendations to apply the CCE method for non-cancer diseases.**Problem:** More data are needed regarding individual compounds in tobacco and TRP emissions to screen for the adverse health effects indicative of non-cancer diseases associated with tobacco emissions other than carcinogenicity, such as cardiovascular diseases, diabetes, chronic obstructive pulmonary disease, and fertility problems.
**In addition to the recommendations listed for carcinogenicity, we add the following recommendations for non-carcinogenic effects:**

•Better use of in vivo data obtained in other legislative frameworks should be made possible to allow their use for other purposes. Disease models are not part of the standard data requirements of chemical legislation, such as REACH, but the in vivo micronucleus test, 90-day toxicity study, the prenatal development study, and the reproductive toxicity/developmental screening study are. Therefore, studies with compounds present in tobacco emissions may be available in the REACH database (International Uniform ChemicaL Information Database, IUCLID) or may become available in the (near) future, although it must be noted that the oral route is preferred within this legislative framework [41], whereas the inhalation route is more relevant for TRP emission exposure;•Substantial efforts are needed to develop and validate in vitro assays for tobacco product-associated non-cancer diseases.


## 4. CCE Method for Multiple Effects

Applying the CCE method to multiple health effects would result in various CCEs, one for each disease (or surrogate toxicological endpoint). In practice, however, this would require data regarding clearly defined diseases, such as atherosclerosis, which is rare, while related endpoints, such as the pathology of the heart and vascular system, or blood cholesterol, usually did not show a significant trend in the studies we reviewed.

As an alternative, the CCE method could be fed by RPFs estimated from available dose–response data, whatever the associated type of effect. Such an approach would result in an integral CCE, irrespective of the specific health effect. This would mean that no conclusions can be drawn regarding specific health outcomes for a TRP associated with the CCE; in fact, the CCE is expected to result in the mitigation of different types of effects in different individuals. For example, some individuals will develop cancer at a later age, while others will develop cardiovascular disease at a later age. Possibly, methodology could be developed to translate the change in this integral CCE into a rough estimate of the mitigation in life years lost.

## 5. Discussion

Various approaches for the risk assessment of complex mixtures were conceptualized by The European Food Safety Authority (EFSA), from pragmatic and simple to refined, each dependent on both the availability of data and the outcome of the risk assessment [13]. Although the approach evaluating combined exposure to multiple compounds in the emissions of TRP and other complex mixtures is essentially the same as one of the methods described by EFSA (i.e., dose addition), the health impact evaluation of tobacco products differs from the usual goal of risk assessment: it aims to evaluate harm reduction claims and to quantify the change in cumulative exposure. For the assessment of the complex mixtures in cigarette or TRP emissions, various concepts have been proposed, ranging from the hazard index (HI) approach [42] or the RPF approach [12], to in vitro or in vivo whole mixture studies [27,43,44]. More pragmatic approaches to evaluating combined exposure to multiple chemicals are the Mixture Assessment Factor (MAF), as discussed in the context of REACH [45], or the threshold of toxicological concern (TTC) approach [46,47].

Currently, in vivo studies with whole smoke are the most inclusive and relevant for estimating health risks from smoking products, because they test the product as a whole in a complex system, such as an animal model [27,44]. However, one test system may be more appropriate for studying the development of a certain disease or detrimental effect than another (see discussion on studying atherosclerosis above). Nevertheless, there is currently no guidance on the selection of a worst-case or most reasonable choice in topography or product type for read-across among products (e.g., such as the Read-Across Assessment Framework (RAAF) under REACH [48]). Both product type and topography determine the presence and quantities of compounds in the emissions, but little information is known on how the emissions are affected. Therefore, information on the health impact of a TRP can only be collected by testing the specific TRP with different topographies. Furthermore, due to the ethical issues involved in performing animal studies to assess the health impact of products such as TRPs and questions regarding the predictability of animal data for human disease, less animal-invasive solutions need to be sought.

Alternatively, the evaluation of RPFs can be conducted based on toxicity studies examining individual compounds, as the advantage is that the dose–response data of these studies can be applied for various compositions, as they may be encountered in practice. Such methods include the CCE method, a promising concept for the evaluation of different TRPs, as it results in a quantitative estimate of reduced cumulative exposure, including the uncertainties in that estimate. The latter is crucial to avoid incorrect conclusions from point estimates where the uncertainties are not made visible. In addition, the CCE method can be used to identify compounds that substantially contribute to cumulative exposure, and hence to the eventual health impact (lifespan). This is especially important for the regulation of products as proposed by the World Health Organization (WHO), with the list of compounds for mandated lowering [18], as it allows the identification of the compounds that substantially contribute to the products’ health risks. The method is very information-intensive, and requires specific data input. However, complex mixtures cannot be evaluated with simple data. The accuracy of the outcome only grows with the availability of reliable data for more compounds.

## 6. Conclusions

In an attempt to refine our CCE method for TRPs we encountered a lack of data on emissions and hazard. This is the basis of our plea for more hazard and emission data for individual compounds occurring in TRPs, and for public access and full data transparency in order to (1) use these data for exposure assessment, (2) increase knowledge of the health effects of the complex mixtures of novel TRP, and (3) evaluate claims regarding reduced harm. This would provide a better understanding of the health impacts associated with TRP without the need to perform additional (animal) tests for complex mixtures, to make the best use of the data that are already available in other legislative frameworks, and to gain information regarding the relative health impacts of TRP use. In the EU, products only need to be notified (and not authorized) before they enter the market. In this notification, only available information on their emissions and their toxicological profiles needs to be submitted to the regulator, without a prescription on how to acquire such data; thus, manufacturers could choose their own study designs. These factors complicate and impede obtaining RPFs to improve and substantiate our CCE method.

Dose–response data of individual compounds should be collected in a database that allows its use for (TRP) health impact evaluation. Missing hazard data should, similar to emission data, be provided by producers. Furthermore, we request researchers to take into consideration our recommendations for future research to transition toward the use of non-animal approaches in mixture risk assessment. In the meantime, the cumulative exposure can be determined with the limited information available, while keeping in mind that the lack of data has an impact on the outcomes. Until more data on emissions and hazards are available, regulators are advised to be very cautious when confronted with a claim for the reduced health impact of TRPs.

## Figures and Tables

**Table 1 ijerph-19-10528-t001:** Inventory of compounds in cigarette smoke (CS) or heated tobacco product (HTP) emissions for which data are available for carcinogenicity classification, emissions, and inhalation unit risks.

Compound	IARC Carcinogenicity Classification	Availability of Emissions Data (CS and HTP)	Availability of Carcinogenicity Data (Inhalation Unit Risk)
Acrylonitrile	2B	√	√
Acetaldehyde *	2B	√	√
1,3-butadiene *	1	√	√
3-monochloro-1,2-propanediol	2B	√ (new)	√ (only oral)
Ethylene oxide	1	√	√
Formaldehyde *	1	√	√
2-furanmethanol	2B	√ (new)	√
Furfural	3	√ (new)	√ (only oral)
Glycidol	2A	√ (new)	√ (only oral or ip)
Benzo[a]pyrene *	1	√	√
Nitrobenzene	2B	√	√
Propylene oxide	2B	√	√
Allyl glycidyl ether	NC		√
Alpha-methyl styrene	2B		√
1,2-dibromo-3-chloropropane	2B		√
1,2-dibromoethane	2A		√
Decalin	NC		√
Hydrazine	2A		√
Isobutyl nitrite	2B		√
Naphthalene	2B		√
Propylene glycol mono-t-butyl ether	2B	√ (new)	√

* World Health Organization (WHO) Study Group on Tobacco Product Regulation (WHO TobReg) list of nine priority compounds in emissions advised for mandated lowering [18]; CS, cigarette smoke; HTP, heated tobacco product; ip, intraperitoneal injection; NC, not classified by IARC; 1 is established carcinogen, 2A is probably carcinogenic, 2B is possibly carcinogenic by IARC, and 3 is unclassifiable according to IARC; (new) means that information on this compound became available after the publication of Slob et al. [12].

## Data Availability

Not applicable, data is included in the paper or in the references cited.

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
