# Peer review of "Challenges in Predicting the Change in the Cumulative Exposure of New Tobacco and Related Products Based on Emissions and Toxicity Dose–Response Data"

_ijerph, 2022, doi:10.3390/ijerph191710528_

Round 1

Reviewer 1 Report

This is an excellent manuscript arguing the case for more data on adverse health effects and emission composition from tobacco products. The authors have developed a cumulative exposure method and it appears to be powerful to utilize the current database to provide meaningful information to substantiate regulatory decisions. I consider this special communication useful for the toxicology community and recommend its publication in the International Journal of Environmental Research and Public Health

Author Response

Thank you very much for your positive review of our manuscript.

Reviewer 2 Report

This is a good perspective/reflection paper on current deficiencies relative to available information for the just assessment of the dangers presented by modern tobacco products. Just a few minor observations to improve the quality of the manuscript even more:

Line 121: acronyms NOAEL, BMD are not explained anywhere in the manuscript

Line 191: same comment regarding the acronym EFSA (although this one is probably well-known)

Overall, there are many acronyms in the paper, so perhaps a centralized list of abbreviations would be a good idea for the quick reference of readers when perusing the manuscript.

There was great care from the authors when writing the body of the manuscript, but the References are formatted very poorly, in a nonunitary fashion, missing journal titles often, "Biochemistry" appears as an author name in one instance and many more mistakes reflecting carelessness when putting together the Bibliography. Therefore, the Bibliography needs a complete tune-up! 

Author Response

Thank you for your review of our manuscript. As you suggest, we have included a list of abbreviations and also checked whether all abbreviations were explained in the text.

We have also checked and corrected the bibliography.